# Sustainable Earnings: How Can Herd Behavior in Financial Accumulation Feed into a Resilient Economic System?

**Aurelie Charles** [1,2,*] and **Damiano Sguotti** [3]

1. Centre for Development Studies, University of Bath, Bath BA2 7AY, UK
2. Centre for the Analysis of Social Policy, University of Bath, Bath BA2 7AY, UK
3. Department of Social & Policy Sciences, University of Bath, Bath BA2 7AY, UK; damiano.sguotti@bath.edu
* Correspondence: a.charles@bath.ac.uk

**Abstract:** The paper applies a methodological tool able to frame national policies with sustainable financial flows between social groups. In effect, exchange entitlement mapping (E-mapping) shows the interdependency of capital and labor earnings across social groups, which is then accounted for in the policy planning of future financial flows for the green transition. First, the paper highlights the extent to which herd behavior feeds into capital and labor earnings by social, occupational, demographic, and regional groups for the United Kingdom, France, and Italy over the past 40 years. Second, learning from these past trends, the paper proposes a policy framing of "sustainable earning trends" to hamper or facilitate financial flows towards sectors, occupations, and regions prone to herd behavior. The paper concludes that for an economic system to be resilient, it should be able to recycle external shocks on group earnings into economic opportunities for the green transition.

**Keywords:** group behavior; financial accumulation; earnings





## 1. Introduction

Since the COP21 agreement in 2015, too little has been done to implement sustainable pathways for environmental policies to meet the target that global warming should not increase by more than 2 °C above pre-industrial levels [1]. Top-down approaches to policy implementation make use of the hierarchy of power relationships in decision-making, with the assumption that the decisions taken by political institutions at the national and global levels necessarily lead to a positive impact on the environment [2]. The argument put forward here is that relying on such vertical relationships of power may trigger rent-seeking behavior in terms of financial accumulation, as seen during the Great Recession [3,4], and may then feed into greenwashing practices. In effect, one dominant feature of excessive capitalism has been the growing hegemony of shareholder value as a mode of governance over human and natural resources [5–7]. At a time of urgent action for financing climate adaptation, such a phenomenon would compromise the intended outcome of an ethical distribution process of financial resources towards a sustainable goal, as understood by Raworth [6], to meet the human needs for all within the planetary boundaries.

In the COP21 era, when global financial flows need to be channeled towards the green transition, there is an urgent need to move our understanding in economic theory and policy from individualism to groupism, in the way that resources are exchanged across economies and societies over time. In this paper, we show that financial accumulation at the individual level in the past was based on group rather than individual behavior. Then, the proposition made here to avoid the negative effects of herd behavior on financial accumulation is that economic policies must be grounded in methodological groupism rather than individualism, which will, in turn, allow future financial flows to be more resilient to external shocks by quickly reaching all parts of society. The main research question raised here is, therefore, "Can private earnings feed into the financial needs of the green transition without feeding into herd behavior that negatively affects the global ecosystem?"

Just as nature thrives on diversity, this paper argues that a resilient economic system based on financial flows free from negative herd behavior in financial accumulation is able to recycle external shocks into economic opportunities within the planetary boundaries [6]. In order to address the main research question, the paper comprises the following two steps. The paper is structured as follows: we start by mapping out group behavior of past capital and labor earnings for the United Kingdom, France, and Italy. In the second part of this paper, we propose the definition of "sustainable earnings trends", whereby financial flows are broken down horizontally by demographic group, and past, present, and potential future scenarios, to serve the purpose of providing transparency on the extent to which financial accumulation by social groups can hamper or facilitate financial flows towards sectors, occupations, and regions prone to herd behavior. We then provide an example of how such a concept can be applied at the national level, using the T21 framework as an example of a policy tool, before providing a few policy recommendations.

## 2. Financial Accumulation: Individual or Group Phenomenon?

In most economics textbooks for Year 1 students, economics departments worldwide teach that individual income is a function of a variety of human capital factors, such as marginal productivity, educational background, skills, and so on [8]. Such methodological individualism means that the discriminatory elements of socialization attached to gender, race, class, or ethnicity are embedded across those individual characteristics and are, as such, not fully accounted for in economic exchange. However, such discriminatory elements describing the power relationships between social groups in a particular context become central to the ways by which income is generated and wealth is accumulated over time.

In behavioral economics, the literature distinguishes between group and individual behavior [9–11], whereby norms of behavior by a social group tend to have an impact on individual decision-making. Similarly, in stratification economics, various authors show how race and ethnic group disparities in market outcomes can be sustained and exacerbated over time [12–15]. In effect, the relative economic value socially assigned to groups of individuals is mostly historically determined and culturally embedded. Social norms convey the rules of legitimacy for the access to resources between social groups, where group membership is sustained according to certain ideal criteria of behavior that sustain group membership. An individual belongs to multiple identities that are shaped by social interactions, and each identity socially entitles him or her to a socially acceptable level of resources [16,17]. Social identities are endogenous to an individual's personal identity since they tend to evolve dynamically over time and thus, questions the use of methodological individualism to capture the dynamics of inequality between social groups [16]. In effect, there is a deterministic element of social life that shapes the way individuals access resources—a perceived legitimacy in social exchanges, which depends on the position of the individual's identities on the spectrum of context-based social stratification [16–18].

When economic exchange takes place, social norms serve as rules for reproducing the advantages of certain social groups at the expense of others. For instance, in the context of the United States, [4] have shown how occupational, race, and gender characteristics are reinforced by the exacerbation of earning differentials between demographic groups during the financialization period of 1980 to 2010. Another example at the intersection of context and educational elites is the evidence from England and Wales that shows that a large number of employers offering the top-paid jobs in the country target an average of only 19 universities (out of 130) in the United Kingdom for those jobs [19]. These examples go beyond the issue of statistical discrimination since group productivity is not responsible for income inequality across all occupations [20]. Rather, the problem lies in the combined effect of identities on inequality since the sum of identities can lead to worse discriminating outcomes than when considering identities separately, as argued by the intersectionality literature [21–23]. Compared with implicit discrimination [24] or with Becker's taste discrimination, the concept of intersectionality departs from methodological

individualism by questioning the boundaries that can be drawn between groups and by defining individuals by a unique combination of diverse groups. As such, it allows us to assess the multiple layers of discrimination over time.

The methodology used to map out group earnings is also known as "exchange entitlement mapping", or "E-mapping" in the literature (see [25,26] on E-mapping theory and its applications in different contexts of analysis in [3,4,27,28]). Such a method allows us to show how social norms are the channels through which the economic environment of individuals affects their opportunities and freedoms to choose different states of well-being [26]. The main concept that is operationalized, similarly to [3,4,26–28], is that income flows between group identities rather than individuals. Such a theoretical standpoint requires that individual income data be aggregated at the group level, from which cointegration analysis can be performed, including unit root testing or Vector Autoregression, to understand the long-term dynamics of income flows towards some groups at the expense of others [3,4,27]. The number of data points from the dataset used below [29], however, did not allow us to perform a full cointegration analysis. Simple Vector Autoregressions by pairs of group earnings were therefore performed as follows [30].

Starting from Charles and Vujic [27], we assumed a society with two demographic groups, $i$ and $j$, both belonging to the same occupational group k. Therefore, individuals are composed of groups $i$ and $k$ or of group identities $j$ and $k$. A socially dominant group is represented by $j$ and receives a premium for group membership, while the non-dominant group is represented by $i$, whose earnings are discriminated against due to group membership. Hence, we assumed a ranking of groups of $j > i$, dependent upon the context specificity in which this ranking has been socially and historically determined.

At the societal level, the sum of earnings from capital and labor $z = \sum(r + w)$ is then distributed between all groups, such that $Z = \sum_{k=0}^{n}(z_i + z_j)$. The point of the model is to show the nature of the short-run relationships of labor and capital earnings between $i$ and $j$, whether the relationship is statistically significant (positively or negatively), or non-significant. In other words, the model describes the share of the capital and labor earnings going towards $i$ and $j$ in $Z$. In the short run, at one end of the individualist spectrum, the first scenario is that capital earnings per group $i$ and $j$ at the occupational level $k$ will depend on the group's productivity and on its earnings in the previous period, calculated as follows:

$$\begin{cases} r_{j(t)} = \alpha + \beta_1 r_{j(t-1)} + \varepsilon_t \\ r_{i(t)} = \alpha + \beta_3 r_{i(t-1)} + \varepsilon_t \end{cases} \tag{1}$$

where capital earnings per demographic group at time $t$ depends on a constant, on its previous value at time $t - 1$, and a white noise term, while labor earnings will be calculated as follows:

$$\begin{cases} w_{j(t)} = \alpha + \beta_1 w_{j(t-1)} + \varepsilon_t \\ w_{i(t)} = \alpha + \beta_3 w_{i(t-1)} + \varepsilon_t \end{cases} \tag{2}$$

where labor earnings per demographic group at time $t$ depends on a constant, on its previous value at time $t - 1$, and a white noise term. Equations (1) and (2) work simultaneously in Z.

To test whether past earning trends have experienced elements of group behavior with a premium attached to group $j$, the following Vector Autoregression analysis is conducted with the following earning relationships:

$$\begin{cases} r_{j(t)} = \alpha + \beta_1 r_{j(t-1)} + \beta_2 r_{i(t-1)} + \varepsilon_t \\ r_{i(t)} = \alpha + \beta_3 r_{i(t-1)} + \beta_4 r_{j(t-1)} + \varepsilon_t \end{cases} \tag{3}$$

where capital earnings per demographic group at time $t$ depends on a constant, on its previous value at time $t-1$, on the value of the other group's earnings at time $t-1$, and a white noise term; while labor earnings will be:

$$\begin{cases} w_{j(t)} = \alpha + \beta_1 w_{j(t-1)} + \beta_2 w_{i(t-1)} + \varepsilon_t \\ w_{i(t)} = \alpha + \beta_3 w_{i(t-1)} + \beta_4 w_{j(t-1)} + \varepsilon_t \end{cases} \tag{4}$$

where labor earnings per demographic group at time $t$ depends on a constant, on its previous value at time $t-1$, on the value of the other group's earnings at time $t-1$, and a white noise term.

At the other end of the spectrum, a second scenario is that, if group membership influences the dynamics of earnings between groups, then Equations (3) and (4) in Z will apply. Here, the nature of the earning relationships between $i$ and $j$ will depend on the sign and statistical significance of $\beta_1$ and $\beta_2$ for (3), and $\beta_3$ and $\beta_4$ for (4). From this perspective, group membership is socially assigned by a dominant convention rather than chosen individually, consciously, or unconsciously, and reproduced over time. While the constant $\alpha$ represents the labor earning gaps between $i$ and $j$ in (4) and the capital earning gaps between $i$ and $j$ in (3), the analysis is more concerned with the short-run dynamics of group biases. The results in Tables 1–3 presented in the next section are therefore concerned with the sign and statistical significance of $\beta_1$ and $\beta_2$ for (3), and $\beta_3$ and $\beta_4$ for (4), while the $\alpha$ gaps are displayed in the Appendix A. Furthermore, as described above, context matters for the ways by which income is generated and wealth is accumulated by a social group over time. Therefore, groups $i$ and $j$ and occupation k will differ across countries. Hence, the empirical testing of Equations (1)–(4) was applied to the United Kingdom, Italy, and France depending on country-dependent classifications.

**Table 1.** Significant relationships of labor and capital earnings between demographic groups in the United Kingdom (1969–2016).

| | Labor Earnings at Time $t$ | | | | | | Capital Earnings at Time $t$ | | | | | |
|---|---|---|---|---|---|---|---|---|---|---|---|---|
| | Male | Female | White | Mixed | Asian | Black | Male | Female | White | Mixed | Asian | Black |
| **Managerial and professional occupations** | | | | | | | | | | | | |
| Male (t-1) | + | + | | | | | + | + | | | | |
| Female (t-1) | . | . | | | | | . | + | | | | |
| White (t-1) | | | -++ | . | . | + | | | -.. | + | . | . |
| Mixed (t-1) | | | + | . | | | | | . | - | | |
| Asian (t-1) | | | . | | + | | | | . | | . | . |
| Black (t-1) | | | . | | | . | | | . | | | |
| **Other skilled occupations** | | | | | | | | | | | | |
| Male (t-1) | + | + | | | | | - | - | | | | |
| Female (t-1) | . | . | | | | | + | + | | | | |
| White (t-1) | | | +++ | + | . | + | | | -.. | - | . | . |
| Mixed (t-1) | | | - | . | | | | | . | + | | |
| Asian (t-1) | | | . | | . | | | | . | | . | |
| Black (t-1) | | | . | | | . | | | . | | | . |
| **Elementary occupations** | | | | | | | | | | | | |
| Male (t-1) | + | + | | | | | . | . | | | | |
| Female (t-1) | - | . | | | | | . | . | | | | |
| White (t-1) | | | +++ | + | . | + | | | +.+ | + | . | . |
| Mixed (t-1) | | | . | - | | | | | . | . | | |
| Asian (t-1) | | | . | | . | | | | + | | . | |
| Black (t-1) | | | . | | | . | | | + | | | + |

**Table 2.** Significant relationships of labor and capital earnings between demographic groups in France (1978–2010).

| | Labor Earnings at Time $t$ | | | | | | | | Capital Earnings at Time $t$ | | | | | | | |
| --- | --- | --- | --- | --- | --- | --- | --- | --- | --- | --- | --- | --- | --- | --- | --- | --- |
| | Male | Female | French | French Naturalized | Non-Citizen | African Citizenship | North-African | European | Male | Female | French | French Naturalized | Non-Citizen | African Citizenship | North-African | European |
| **Managerial and professional occupations** | | | | | | | | | | | | | | | | |
| Male (t-1) | . | . | | | | | | | . | . | | | | | | |
| Female (t-1) | + | + | | | | | | | + | + | | | | | | |
| French (t-1) | | | +++++ | + | - | + | + | + | | | -+-+ | - | | - | + | + |
| Nat.F (t-1) | | | - | - | | | | | | | + | - | | | | |
| Non-cit (t-1) | | | - | | - | | | | | | | | | | | |
| African (t-1) | | | + | | | - | | | | | + | | | - | | |
| N-Afr (t-1) | | | . | | | | . | | | | - | | | | + | |
| Europ. (t-1) | | | . | | | | | . | | | - | | | | | . |
| **Other skilled occupations** | | | | | | | | | | | | | | | | |
| Male (t-1) | . | + | | | | | | | - | . | | | | | | |
| Female (t-1) | . | . | | | | | | | + | + | | | | | | |
| French (t-1) | | | - ++++ | - | + | . | . | . | | | - - ++ | + | | - | . | + |
| Nat.F (t-1) | | | + | + | | | | | | | - | + | | | | |
| Non-cit (t-1) | | | + | | - | | | | | | | | | | | |
| African (t-1) | | | . | | | . | | | | | + | | | - | | |
| N-Afr (t-1) | | | . | | | | . | | | | . | | | | . | |
| Europ. (t-1) | | | . | | | | | . | | | . | | | | | - |
| **Elementary occupations** | | | | | | | | | | | | | | | | |
| Male (t-1) | . | . | | | | | | | . | . | | | | | | |
| Female (t-1) | . | . | | | | | | | + | + | | | | | | |
| French (t-1) | | | ++.+. | + | + | . | . | . | | | -+- | . | | - | + | . |
| Nat.F (t-1) | | | + | . | | | | | | | + | + | | | | |
| Non-cit (t-1) | | | . | | . | | | | | | | | | | | |
| African (t-1) | | | . | | | . | | | | | + | | | - | | |
| N-Afr (t-1) | | | . | | | | . | | | | - | | | | + | |
| Europ. (t-1) | | | . | | | | | . | | | . | | | | | . |

**Table 3.** Significant relationships of labor and capital earnings between demographic groups in Italy (1986–2016).

| | Labor Earnings at Time $t$ | | | | Capital Earnings at Time $t$ | | | |
| --- | --- | --- | --- | --- | --- | --- | --- | --- |
| | Male | Female | Born in | Born out | Male | Female | Born in | Born out |
| **Blue-collar occupations** | | | | | | | | |
| Male (t-1) | + | + | | | + | + | | |
| Female (t-1) | . | . | | | . | + | | |
| Born in (t-1) | | | . | . | | | . | . |
| Born out(t-1) | | | . | . | | | . | . |
| **Office workers and school teachers** | | | | | | | | |
| Male (t-1) | . | . | | | . | - | | |
| Female (t-1) | . | + | | | + | + | | |
| Born in (t-1) | | | . | . | | | . | . |
| Born out(t-1) | | | . | . | | | . | . |

| | Labor Earnings at Time *t* | | | | Capital Earnings at Time *t* | | | |
|---|---|---|---|---|---|---|---|---|
| | Male | Female | Born in | Born out | Male | Female | Born in | Born out |
| **Junior/middle managers and professional occupations** | | | | | | | | |
| Male (t-1) | + | + | | | + | . | | |
| Female (t-1) | . | . | | | . | . | | |
| Born in (t-1) | | | + | + | | | . | . |
| Born out(t-1) | | | . | - | | | . | + |
| **Senior managers and white-collar workers** | | | | | | | | |
| Male (t-1) | . | . | | | . | . | | |
| Female (t-1) | . | . | | | . | . | | |
| Born in (t-1) | | | + | + | | | . | . |
| Born out(t-1) | | | . | - | | | - | . |

## 3. Trends in Capital and Labor Earnings in the United Kingdom, Italy, and France

The accumulation of earning excesses in the financial sector is now widely recognized to be one of the features of the evolution of income distribution over the past century [7,31]. One potential explanation put forward by Piketty and Saez [31] is the role of norms in exacerbating earnings at the top of the income distribution. In effect, group behavior at the top of managerial and financial occupations has been an essential factor that has led to financial excesses. This section presents the trends of the horizontal income inequalities over time, referring to three different geographical contexts—the United Kingdom, France, and Italy—showing the dominant relationships in financial accumulation.

In terms of methodology, for the United Kingdom, France, and Italy, individual data on labor and capital earnings were aggregated at the group level, including gender, class, immigrant status, and other demographic variables depending on the country's classifications (LIS (2020)). LIS (2020) is a harmonized database of microdata on income and other economic personal and household variables. Labor and capital earning variables were plotted for each country and group, below, to test Equations (1)–(4) for each country. VARs allowed us to generalize the dependencies of two variables over time, as displayed in Equations (3) and (4), using LIS data (2020) [29,30]. Tables 1–3 below are simple visual representations of the VARs shown in the Appendix A, as they display only the signs of the statistically significant coefficients $\beta_1$ to $\beta_4$ in Equations (1)–(4) of the model described above—the dots representing the non-significant coefficients. The limitation of the data-driven methodology is essential in terms of the various combinations of relationships that can be inferred. Hence, prior knowledge of the historical relationships between demographic groups, together with the availability of data representing these groups, is crucial for the validity of the model.

### 3.1. Capital and Labor Earnings in the United Kingdom (1994–2016)

In the United Kingdom, the breakdown of the capital and labor earnings based on LIS data [29] are by gender and ethnicity (white, mixed race, Asian, and Black), with three main occupations (managers and professionals, other skilled workers, and laborers/elementary), as displayed in Table A1 of the Appendix A. Figure 1 below represents the average labor income for the white and Black groups from 1994 to 2016. While the Black trend starts to overtake the white trend in the mid-2000s, the Great Recession brought the trends back to their "normal" dominant–dominated relationship.

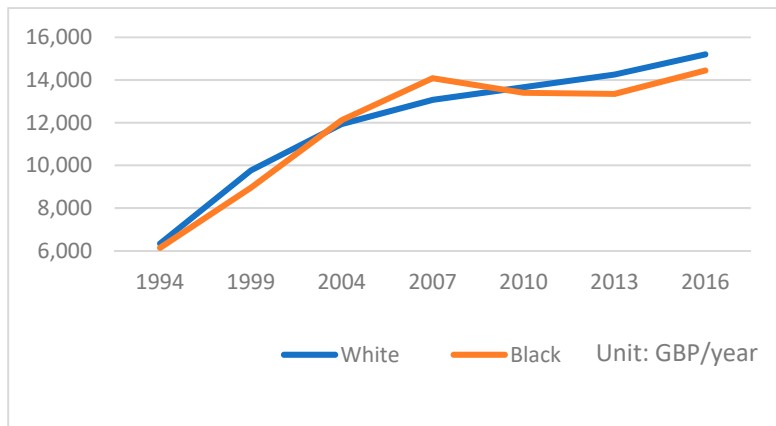

**Figure 1.** Average labor income by ethnicity in the United Kingdom (1994–2016). Source: Authors' elaboration from LIS data (2020).

Figure 2 shows the average of all cash payments from property and capital (including financial and non-financial assets—interest and dividends, rental income, and royalties) for the Black and white groups. The evolution of the existing gap is quite significant in describing the relationship of power between the two groups in financial accumulation. While the white group benefited from a sharp increase in capital income in the build-up towards the Great Recession, the trend for the Black group was stagnant in the same period, followed by an increasing capital income gap. Here, again, the interdependence of group earnings over time is striking.

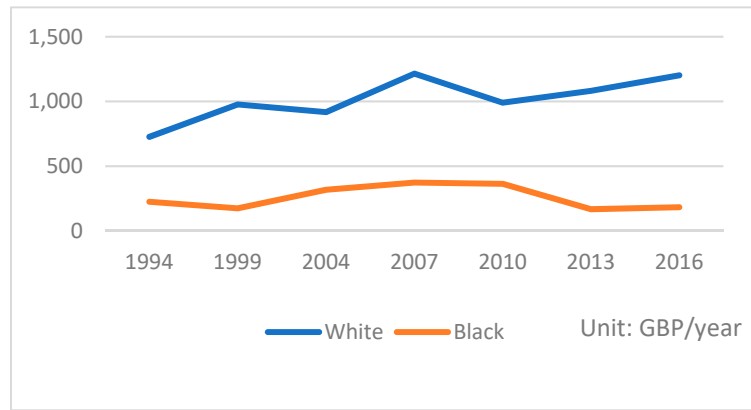

**Figure 2.** Average capital income by ethnicity in the United Kingdom (1994–2016). Source: Authors' elaboration from LIS data (2020).

Looking at the VARs in Table A1 of the Appendix A, numerous coefficients $\beta_1$ to $\beta_4$ across occupations are statistically significant at the 5 or 10% level. This points towards an interdependency of capital and labor earnings between groups at the occupational level, which is more visible in Table 1, below, reporting the signs of the statistically significant coefficients from the VARs of Table A1.

The relationships of labor and capital earnings displayed in Table 1 would reflect Equations (1) and (2) if all signs in the grey diagonals were positive and none of the other cases were positive or negative. Instead, we see a pattern emerging here whereby, on the one hand, the male and white group's labor earnings at time $t-1$ are positively correlated with female labor earnings at time $t$, and with the labor earnings of the mixed and Black groups across all three occupations. On the other hand, labor earnings at $t-1$ of the minority groups tend not to be significant with their own earnings at time $t$. For capital earnings, statistical significance is not as widespread, given the extent of the wealth gaps between the dominant and minority groups. It is worth noting, however, that there is a

strong interdependence of earnings in the elementary occupations with the capital earnings of minority groups at time $t - 1$ being positively related to the capital earnings of the white group at time $t$.

### 3.2. Capital and Labor Earnings in France (1978–2010)

In France, the breakdown of the capital and labor earnings based on LIS data (2020) are by gender and citizenship (French, French-naturalized, non-citizen, African, Northern African, European, and others), with three main occupations (managers and professionals, other skilled workers, and laborers/elementary), as displayed in Table A2 of the Appendix A.

Figures 3 and 4 represent the trends of labor and capital income by class from 1978 to 2010 in France. The trends of labor income show an increasing gap since the 1990s between white-collar and blue-collar workers, while skilled workers experienced trend-stationary labor income over the period. For the trends of capital income, there is a similar increasing gap between white-collar workers and the other two groups, and all three trends were hit by the Great Recession of 2008. Overall, such data shows that, regardless of skills, productivity, or ethnic background, the rising gap shows that there is a pattern of a dominating–dominated relationship, horizontally, between white-collar workers and the other two categories.

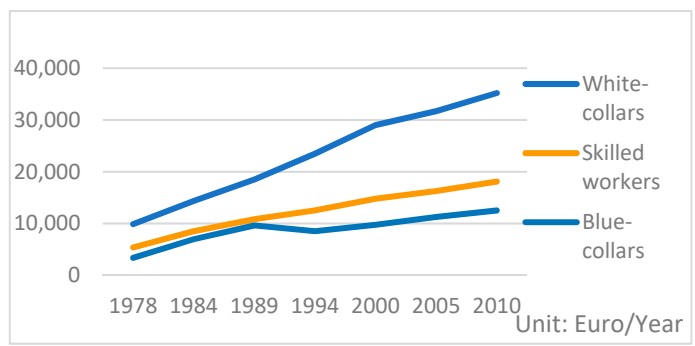

**Figure 3.** Average labor income by class in France. Source: Authors' Elaboration form LIS data (2020).

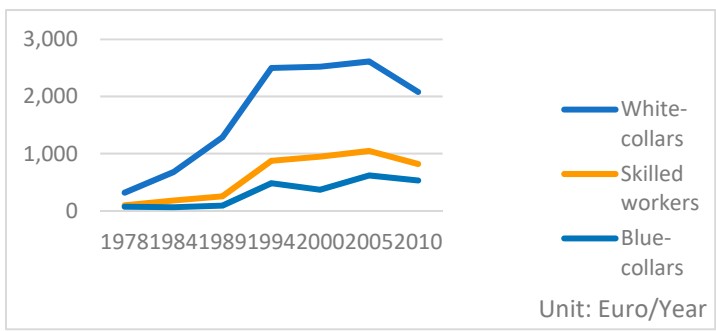

**Figure 4.** Average capital income by class in France. Source: Authors' Elaboration from LIS data (2020).

Looking at the VARs in Table A2 of the Appendix A, numerous coefficients $\beta_1$ to $\beta_4$ across occupations are statistically significant at the 5 or 10% level. This points towards an interdependency of capital and labor earnings between groups at the occupational level, which is more visible in Table 2, below, reporting the signs of the statistically significant coefficients from the VARs of Table A2.

The relationships of labor and capital earnings displayed in Table 2 would again reflect Equations (1) and (2) if all signs in the grey diagonals were positive and none of the other cases were positive or negative. Instead, there is a similar pattern of interdependence between the dominant French group and minority groups for labor earnings, whereby the labor earnings of French nationals are correlated significantly to the earnings of minority groups, especially in managerial and professional occupations. Meanwhile, the gender

element of group behavior shows in the labor earnings of men in managerial and professional occupations, and in the capital earnings across occupations, whereby the earnings of women at time $t - 1$ are positively correlated to male earnings at time $t$.

### 3.3. Capital and Labor Earnings in Italy (1989–2016)

In Italy, the breakdown of the capital and labor earnings based on LIS data (2020) are by gender and place of birth (in or out of Italy), with four main occupations (blue-collar workers, office worker and schoolteacher, junior/middle managers and professionals, senior managers and white-collar workers), as displayed in Table A3 of the Appendix A.

Figure 5 shows an increasing gap between white-collar workers and other occupations. By contrast, the trends of capital income by class in Figure 6 do not display a similar increasing gap. Class is only one of the numerous relationships of inequality between social groups in Italy. Well-documented in the literature, it spans from geographical inequality with the North–South divide, in terms of economic development, to gender, immigration status, and age [32–34]. For example, the hourly pay gaps over the past five years have reproduced themselves across those different groups: women earning 7.4% less than men on average, immigrants earning 17.4% less than an Italian citizen, and a young adult between 15 and 29 years old earning 24.2% less than an adult in her/his working life [35].

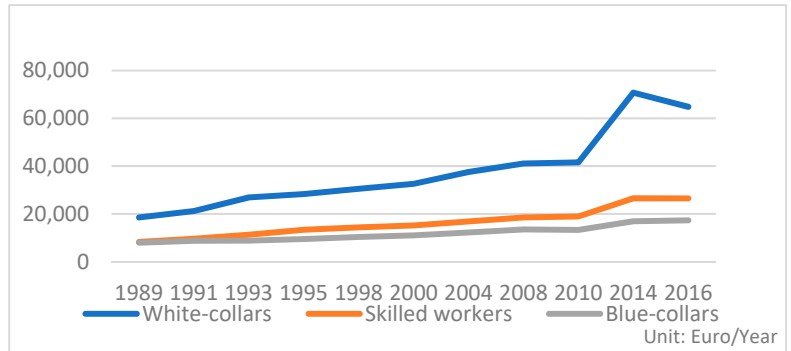

**Figure 5.** Average labor income by class in Italy. Source: Authors' Elaboration from LIS data (2020).

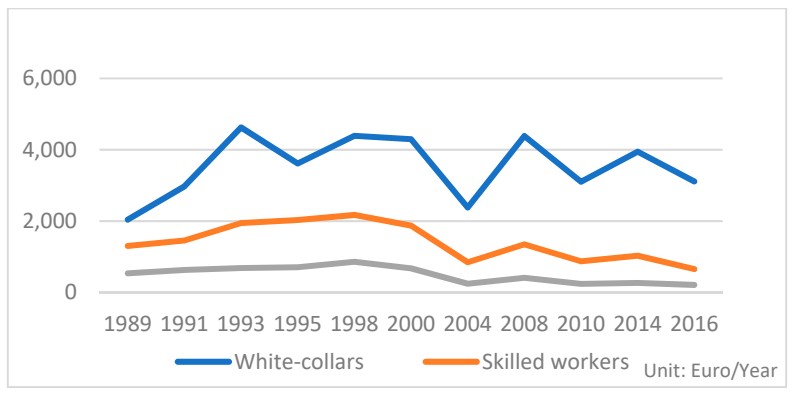

**Figure 6.** Average capital income by class in Italy. Source: Authors' Elaboration from LIS data (2020).

Looking at the VARs in Table A3 of the Appendix A, numerous coefficients $\beta_1$ to $\beta_4$ across occupations are statistically significant at the 5 or 10% level. This points towards an interdependency of capital and labor earnings between groups at the occupational level, which is more visible in Table 3, below, reporting the signs of the statistically significant coefficients from the VARs of Table A3.

The relationships of labor and capital earnings displayed in Table 3 would again reflect Equations (1) and (2) if all signs in the grey diagonals were positive and none of the other cases were positive or negative. Instead, the interdependency of capital and labor earnings

between the male and female groups perpetuates over time, especially in the blue-collar occupations and in the junior/middle managers and professional occupations. Capital earnings of the occupations of office workers and schoolteachers show a strong relationship on a gender basis but not in managerial occupations. The interdependency is stronger for gender than for class, similarly to the United Kingdom.

Overall, the results across the three countries show that for some occupations, there is a strong interdependence of earnings, pointing toward Models (3) and (4), and the results reflecting more in Models (1) and (2) tend to be for male earnings, for the dominant incumbent group (white or nationals), and for labor earnings rather than capital earnings. Some social groups earn more at the expense of others, which is based on social relationships of power more than on individual productivity. There is, therefore, the evidence here of the interdependency of earnings between social groups, whether it is in the United Kingdom, France, or Italy, with varying degrees of interdependency across groups and countries. Such discrepancies are not based on individual productivity but on group biases, whereby some groups are deemed socially and, therefore, economically more valuable than others. Over time, such discrepancies are economically unsustainable, feeding into herd behavior that exacerbates group status and eventually creating production, consumption, and financially speculative bubbles that sustain the social status of the dominant group, making the entire economic system unsustainable. Another methodological perspective on financial accumulation is therefore needed for a sustainable economic system.

## 4. Sustainable Earnings Trends: A Proposition

The proposition made here is that the diversity of group relationships is key to the stability of the global economic system. Looking at Figure 7 as a global network of financial flows, each node represents a social group linking these financial flows. With social groups rather than individuals at the center of financial interactions it allows us to frame the social entitlement rules of financial flows. With each individual in the system belonging to different social groups, any external shock on one of the nodes in the system will be offset by other nodes around. "Efficiency occurs when a system streamlines and simplifies its resource flow to achieve its aims, say by channeling resources directly between the larger nodes. Resilience, however, depends upon diversity and redundancy in the network, which means that there are ample alternative connections and options in times of shock and change." ([6], p. 175).

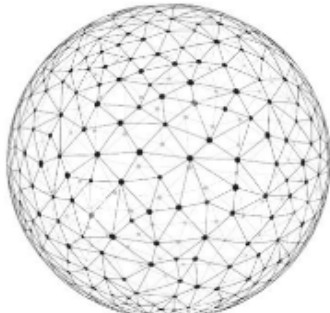

**Figure 7.** A network of flows: structuring an economy as a distributed network can more equitably distribute the income and wealth that it generates. Source: ([6], p. 174).

Building policy tools able to map out group earnings from the past are able to inform future policies of the potential cognitive biases brought about by group behavior in individual decision-making at the micro-level, and are aware of the way such a phenomenon aggregates at the macro-level in financial flows. The approach of E-mapping applied to the United Kingdom, France, and Italy in the previous section shows that social norms are the channels through which the economic environment of individuals affects their earning opportunities. Adopting such a lens could create sustainable earning trends whereby financial flows are broken down horizontally by demographic group, and past, present,

and potential future scenarios, to serve the purpose of providing transparency on the extent to which financial accumulation by social groups can hamper or facilitate financial flows towards sectors, occupations, and regions prone to herd behavior.

The analysis presented above shows the extent to which group behavior overtakes individual motives in financial accumulation, and that the norms of dominant groups guide financial flows across the economy and society. This is especially visible in the financial sector [3,4] due to the magnitude of the flows in that sector, but in light of these above results, it is a phenomenon consistent across the labor force and, hence, society. Such a wide phenomenon questions whether group earnings can ever become "sustainable" earning trends that feed into the green transition of the economy and society. "Sustainability" here relates to maintaining the human biodiversity in society by sustaining the livelihoods of all groups rather than letting financial flows freely float towards one dominant group. In a COP21 era with doom prospects for demand-led growth, if one accepts methodological groupism where groups can be broadly defined in social identity terms (e.g., occupational, geographical, racial, gender, and so on), it is unlikely that individual earnings will feed into the financial needs of the green transition without feeding into group biases and associated financial bubbles. Financial decisions are, in effect, not just for entrepreneurs but also reflect daily consumption, saving, investment, education, and migration decisions made by all individuals. In particular, if group behavior overtakes individual decision-making, it makes us wonder how daily financial decisions, such as personal consumption, savings, investment, as well as education and migration choices can serve the financial needs of the green transition.

Looking at the role of animal spirit in terms of financial decisions, that is "our innate urge to activity" ([36], p. 163 in [37], p. 7), [37] rightly points out the importance of group membership as well as context in influencing individual decision-making. Then, given that individuals belong to multiple groups, the boundaries of which are socially determined, conventions set by salient groups appear, over time, as the rules of the game in financial interactions. As such, in the response to [37]'s argument, [38] clearly spells out that conventions are the actual "context" in which financial interactions take place. Thus, if a context is shaped by group relationships, it raises the question of individual versus group legitimacy in financial flows whereby group norms rather than individual instincts serve as a basis for financial exchanges; in which case a financial bubble similar to the one leading to the 2007–2008 crisis is likely to feed into the green transition. For instance, the share of the financial sector has increased by nearly a third from 2014 to 2015, and if such a trend takes momentum, with 67% of climate-aligned bonds going to transport and 1% to waste and pollution control in 2016 [39], it is likely that the car industry will flourish in a toxic habitat in the coming decades.

To address the legitimacy issue in future financial flows, the methodological proposition of groupism made here is that the legitimacy of financial flows should be first acknowledged to be group-based rather than individual-based, to be able to think in terms of ecological legitimacy in financial flows. Building on the phenomenon of herd behavior in financial decision-making, the following proposition of "sustainable earning trends" is anchored in the rationale that group behavior has more than a speculative impact on financial flows and serves as a basis for financial capacity-building scenarios to finance the green transition. However, as we will now show, reasoning in terms of "sustainable" trends of earnings means that there is an awareness of these group phenomena at the individual and policy levels, which would be the first step to move from social-based to ecological-based legitimacy of financial resources.

### 4.1. Policy Example: The T21 Framework with Sustainable Earnings

Applying the lens of group mapping to any policy tool brings transparency and legitimacy to the policy process and resilience to the economic system impacted by the policy process. In effect, the innovative core of such a lens is to show how social entitlement rules to resources and financial flows, in particular, can become ecological entitlement

rules to a thriving environment. To do so, E-mapping is here applied to one of the current models of development planning, namely the Threshold 21 framework [40]. The Threshold 21 model (T21) is a development planning tool used by national governments to address the financial challenges of turning green by designing business-as-usual and green capacity-building scenarios between sectors for low or decarbonized development and natural resource efficiency. The T21 model suggests that, on average, 1 to 2.5% of global GDP per year is needed up to 2050 to green the economy. The T21 model is based on the existing interconnections between the economic, social, and environmental dimensions of development for a country, hence supporting the idea of sustainable development (see Figure 8). Several empirical applications have already been done in countries as varied as Denmark, China, and Bangladesh. Green scenarios are simulated and compared with business-as-usual, resource-intensive growth, and fossil fuel consumption scenarios. The simulations illustrate that green scenarios are more efficient in achieving environmental targets than all business-as-usual scenarios used in the model. In effect, although during the initial stage of their implementation, green scenarios do not show outstanding results compared with business-as-usual scenarios, in the middle to long-term stages, green scenarios outperform business-as-usual ones for GDP growth.

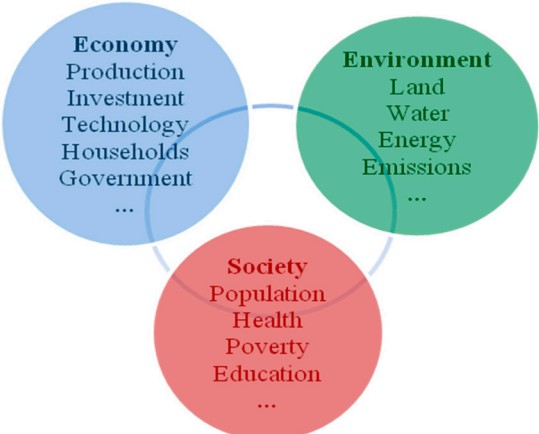

**Figure 8.** Spheres and sectors in the T21 framework. Source: (Tcherneva, 2020).

By adding the group dimension to the T21 framework, as shown in Figure 9 with occupational, demographic, and regional earnings, social entitlement rules become ecological rules of entitlement whereby past cognitive biases brought about by group behavior are mapped out according to a country's specific context, as shown in the previous sections for the United Kingdom, France, and Italy. Such an exercise of group mapping can then inform the dynamics of future financial flows where, with each individual in the system belonging to different social groups (Figure 7), any external shock on one of the nodes in the system will be offset by other nodes around.

*4.2. Policy Recommendations*

Policy instruments based on methodological groupism could be readily applied to inform the future dynamics of financial flows. However, group membership needs first to be acknowledged and accounted for in national statistics. As seen in the three countries covered above, the variables proposed in national statistics do not necessarily cover a key element of discrimination in financial accumulation, which is skin color. In effect, [41] have shown that earning discrimination based on skin color in the United States is passed down through generations. With variables based on citizenship (e.g., French sample) or place of birth (e.g., Italian sample), the discrimination from second-generation migrants and onwards cannot be accounted for, and thus, remains invisible for policymaking.

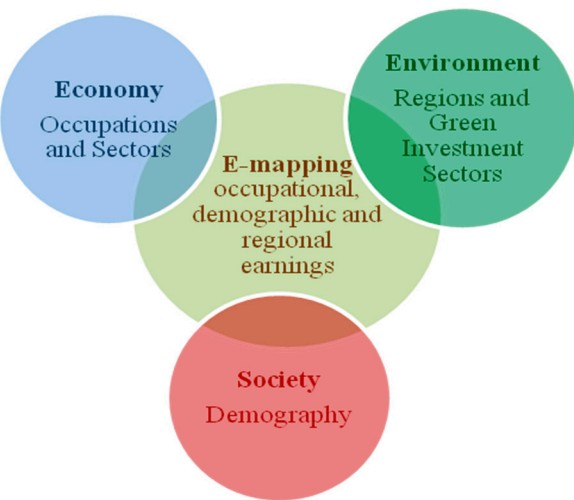

**Figure 9.** T21 methodology with E-mapping. Source: Author's elaboration.

Recommendation 1: Collecting detailed information on the demographics of capital and labor earnings in national statistical institutions in order to provide transparency on the historical trends of financial accumulation.

Secondly, with accurate demographic data based on groups rather than individuals, it becomes straightforward to build (1) business-as-usual earning trends, reproducing a past dynamic of earnings, including potential bubbles of financial accumulation by sector, occupation, and/or demographic group, and (2) sustainable earning trends to direct earnings to the occupations and groups needed for future investments. An open platform of knowledge, mapping the past and potential future scenarios of earnings (1 or 2), could indeed enable individuals, including policymakers, investors, workers, managers, students, men, and women, to inform daily financial choices, the impact past financial choices had on financial flows, and the potential impact current financial decisions can have according to Scenario 1 or 2.

Recommendation 2: Creating an open platform of business-as-usual earning trends and sustainable earning trends by sectoral, occupational, geographic, and demographic groups to inform future sustainable trends of capital and labor earnings.

Finally, the use of this platform of knowledge by individuals can then shape future entitlement rules to financial flows with all its complexity and uncertainty. From a top-down perspective, such a platform could inform policymakers on labor policies (e.g., a job guarantee [42] by demographic group), investment policies (e.g., targeting occupations and sectors supporting the job guarantee), and income and wealth policies (e.g., baby bonds [43] and inheritance tax [44] by demographic group). From a bottom-up perspective, such a platform could inform workers, managers, pensioners, students, men, and women, on the likely impact their daily financial choices have on jobs, savings, and investments by comparing business-as-usual earning choices and sustainable earning choices. There is here no deterministic nature for future flows, rather the future entitlement rules depend on the outcome of individual decisions whether one chooses to follow his or her own groups' behavior or not.

## 5. Conclusions

The paper shows that there is ample evidence of the interdependency of earnings between social groups, whether it is in the United Kingdom, France, or Italy, with varying degrees of interdependency across groups and countries. Such discrepancies are not based on individual productivity but on the social perception that one group is socially and, therefore, economically more valuable than another. Over time, such discrepancies are economically unsustainable, feeding into herd behavior that exacerbates group status

and eventually creates production, consumption, and financially speculative bubbles that sustain that group's status. Another perspective on income accumulation is therefore needed for a sustainable economic system.

National and international agencies that have developed rationales and policy plans to address the climate emergency are based on methodological individualism. Trillions of dollars have been released to "green" the economy. However, this paper shows that these efforts need to account for herd behavior in financial flows. Income and wealth inequalities represent power relationships among social groups, which then set social entitlement rules in economic exchange. Building planning tools at the national and international levels with a group mapping perspective can inform future policies of the potential cognitive biases at the individual level that aggregate at the macro-level. Adopting such a lens could create sustainable earning trends whereby financial flows are broken down horizontally by demographic group to provide transparency on the extent to which capital and labor earnings by social group can hamper or facilitate financial flows towards sectors, occupations, and regions prone to herd behavior.

**Author Contributions:** Conceptualization, A.C.; investigation, A.C.; data curation, D.S.; formal analysis, D.S.; methodology, A.C.; supervision, A.C.; writing—original draft, A.C. All authors have read and agreed to the published version of the manuscript.

**Funding:** This research received no external funding.

**Institutional Review Board Statement:** Not applicable.

**Informed Consent Statement:** Not applicable.

**Data Availability Statement:** Luxembourg Income Study (LIS) Database. Available online: http://www.lisdatacenter.org (accessed on 22 March 2021).

**Conflicts of Interest:** The authors declare no conflict of interest.

## Appendix A

**Table A1.** VARs of capital and labor earnings by occupation, gender, and ethnicity in the United Kingdom (12 Waves 1969–2016). Author's calculation from LIS data (2020).

| Variables Names | Oc1: Managers and Professionals (ISCO 1 & 2) Oc2: Other Skilled Workers (ISCO 3-8, 10) Oc3: Laborers/Elementary (ISCO 9) lab: Average Labor Income cap: Average Capital Income m: Male Population f: Female Population eth1: White Ethnic Group eth2: Mixed Race/Multiple Ethnic Groups eth3: ASIAN/ASIAN BRITISH ETHNIC GROUP eth4: BLACK/AFRICAN/CARIBBEAN/BLACK BRITISH ETHNIC GROUP ** Significance ≤ 0.05 * Significance between 0.10 and 0.05 |
|---|---|
| **Variables** | **VARs** |
| oc1_mlab oc1_flab | Oc1_mlab = 6802.64 * + 0.91 oc1_mlab $(t-1)$ − 0.01 oc1_flab $(t-1)$ Oc1_flab = 2763.66 ** + 0.75 oc1_mlab $(t-1)$ − 0.12 oc1_flab $(t-1)$ |
| oc1_mcap oc1_fcap | Oc1_mcap = 836.92 * + 1.36 oc1_mcap $(t-1)$ − 1.01 oc1_fcap $(t-1)$ Oc1_fcap = 762.79 * +0.70 oc1_mcap $(t-1)$ − 0.40 oc1_oc1_fcap $(t-1)$ |
| oc1_eth1lab oc1_eth2lab | Oc1_eth1lab = 60071.24 * − 0.77 * oc1_eth1lab $(t-1)$ + 0.25 * oc1_eth2lab $(t-1)$ Oc1_eth2lab = 45526.60 − 0.52 oc1_eth1lab $(t-1)$ + 0.36 oc1_eth2lab $(t-1)$ |
| oc1_eth1cap oc1_eth2cap | Oc1_eth1cap = 2930.56 * − 0.60 * oc1_eth1cap $(t-1)$ − 0.17 oc1_eth2cap $(t-1)$ Oc1_eth2cap = −691.47 * + 0.85 * oc1_eth1 cap $(t-1)$ − 0.01 oc1_eth2cap $(t-1)$ |
| oc1_eth1lab oc1_eth3lab | Oc1_eth1lab = 18249.62 * + 0.53 * oc1_eth1lab $(t-1)$ + 0.04 oc1_eth3lab $(t-1)$ Oc1_eth3lab = −12984.76 + 0.79 oc1_eth1lab $(t-1)$ + 0.55 ** oc1_eth3lab $(t-1)$ |
| oc1_eth1cap oc1_eth3cap | Oc1_eth1cap = 1758.42 * − 0.10 oc1_eth1cap $(t-1)$ + 0.16 oc1_eth3cap $(t-1)$ Oc1_eth3cap = 494.53 − 0.37 oc1_eth1cap $(t-1)$ + 1.44 oc1_eth3cap $(t-1)$ |
| oc1_eth1lab oc1_eth4lab | Oc1_eth1lab = 18040.16 * + 0.54 * oc1_eth1lab $(t-1)$ +0.02 oc1_eth4lab $(t-1)$ Oc4_eth4lab = − 21267.1 + 1.71 * oc1_eth1lab $(t-1)$ − 0.24 oc1_eth4lab $(t-1)$ |

**Table A1.** *Cont.*

| Variables | VARs |
|---|---|
| **Variables Names** | Oc1: Managers and Professionals (ISCO 1 & 2)<br>Oc2: Other Skilled Workers (ISCO 3-8, 10)<br>Oc3: Laborers/Elementary (ISCO 9)<br>lab: Average Labor Income<br>cap: Average Capital Income<br>m: Male Population<br>f: Female Population<br>eth1: White Ethnic Group<br>eth2: Mixed Race/Multiple Ethnic Groups<br>eth3: ASIAN/ASIAN BRITISH ETHNIC GROUP<br>eth4: BLACK/AFRICAN/CARIBBEAN/BLACK BRITISH ETHNIC GROUP<br>** Significance $\leq$ 0.05<br>* Significance between 0.10 and 0.05 |

| Variables | VARs |
|---|---|
| oc1_eth1cap<br>oc1_eth4cap | Oc1_eth1cap = 1749.47 * − 0.02 oc1_eth1cap $(t-1)$ − 0.04 oc1_eth4cap $(t-1)$<br>Oc1_eth4cap = −621.16 +0.67 oc1_eth1cap $(t-1)$ + 0.01 oc1_eth4cap $(t-1)$ |
| oc2_mlab<br>oc2_flab | Oc2_mlab = 3155.20 * + 1.25 * oc2_mlab $(t-1)$ − 0.49 oc2_flab $(t-1)$<br>Oc2_flab = 1113.26 * + 0.72 * oc2_mlab $(t-1)$ − 0.11 oc2_flab $(t-1)$ |
| oc2_mcap<br>oc2_fcap | Oc2_mcap = 223.20 * − 4.11 * oc2_mcap $(t-1)$ + 4.29 * oc2_fcap $(t-1)$<br>Oc2_fcap = 241.35 * 2212 3.47 * oc2_mcap $(t-1)$ + 3.83 * oc2_fcap $(t-1)$ |
| oc2_eth1lab<br>oc2_eth2lab | Oc2_eth1lab = −3702.57 * +1.50 * oc2_eth1lab $(t-1)$ − 0.27 * oc2_eth2lab $(t-1)$<br>oc2_eth2lab = −18432.94 + 2.75 ** oc2_eth1lab $(t-1)$ − 0.80 oc2_eth2lab $(t-1)$ |
| oc2_eth1cap<br>oc2_eth2cap | Oc2_eth1cap = 1160.31 * − 0.48 * oc2_eth1cap $(t-1)$ − 0.01 oc2_eth2cap $(t-1)$<br>Oc2_eth2cap = 2745.67 * − 3.43 * oc2_eth1cap $(t-1)$ +0.50 * oc2_eth2cap $(t-1)$ |
| oc2_eth1lab<br>oc2_eth3lab | Oc2_eth1lab = 7065.82 * + 0.68 * oc2_eth1lab $(t-1)$ + 0.01 oc2_eth3lab $(t-1)$<br>Oc2_eth3lab = −2555.72 + 0.77 oc2_eth1lab $(t-1)$ + 0.25 oc2_eth3lab $(t-1)$ |
| oc2_eth1cap<br>oc2_eth3cap | Oc2_eth1cap = 362.58 + 0.32 oc2_eth1cap $(t-1)$ + 0.27 oc2_eth3cap $(t-1)$<br>Oc2_eth3cap = −209.81 + 0.76 oc2_eth1cap $(t-1)$ + 0.85 oc2_eth3cap $(t-1)$ |
| oc2_eth1lab<br>oc2_eth4lab | Oc2_eth1lab = 6707.39 * + 0.76 * oc2_eth1lab $(t-1)$ − 0.05 oc2_eth4lab $(t-1)$<br>Oc2_eth4lab = 3224.54 + 0.84 ** oc2_eth1lab $(t-1)$ − 0.01 oc2_eth4lab $(t-1)$ |
| oc2_eth1cap<br>oc2_eth4cap | Oc2_eth1cap = 617.83 * + 0.24 oc2_eth1cap $(t-1)$ − 0.25 oc2_eth4cap $(t-1)$<br>Oc2_eth4cap = 185.54 + 0.09 oc2_eth1cap $(t-1)$ − 0.45 oc2_eth4cap $(t-1)$ |
| oc3_mlab<br>oc3_flab | Oc3_mlab = 1958.92 * + 1.30 * oc3_mlab $(t-1)$ − 0.73 ** oc3_flab $(t-1)$<br>Oc3_flab = 414.75 + 0.41 * oc3_mlab $(t-1)$ + 0.22 oc3_flab $(t-1)$ |
| oc3_mcap<br>oc3_fcap | Oc3_mcap = 261.07 * + 0.60 oc3_mcap $(t-1)$ − 0.47 oc3_fcap $(t-1)$<br>Oc3_fcap = 171.71 * + 0.36 oc3_mcap $(t-1)$– 0.10 oc3_fcap $(t-1)$ |
| oc3_eth1lab<br>oc3_eth2lab | Oc3_eth1lab = 6944.97 * + 0.47 * oc3_eth1lab $(t-1)$ − 0.02 oc3_eth2lab $(t-1)$<br>Oc3_eth2lab = −7280.15 + 2.17 * oc3_eth1lab $(t-1)$ − 0.71 * oc3_eth2lab $(t-1)$ |
| oc3_eth1cap<br>oc3_eth2cap | Oc3_eth1cap = −333.46 * + 1.88 * oc3_eth1cap $(t-1)$ − 0.03 oc3_eth2cap $(t-1)$<br>Oc3_eth2cap = −742.18 * + 2.42 * oc3_eth1cap $(t-1)$ − 0.25 ** oc3_eth2cap $(t-1)$ |
| oc3_eth1lab<br>oc3_eth3lab | Oc3_eth1lab = 4546.80 * + 0.64 * oc3_eth1lab $(t-1)$ + 0.03 oc3_eth3lab $(t-1)$<br>Oc3_eth3lab = −1565.27 + 1.24 oc3_eth1lab $(t-1)$ − 0.12 oc3_eth3lab $(t-1)$ |
| oc3_eth1cap<br>oc3_eth3cap | Oc3_eth1cap = 80.42 + 0.53 oc3_eth1cap $(t-1)$ + 0.14 * oc3_eth3cap $(t-1)$<br>Oc3_eth3cap = 1657.60 ** − 4.40 oc3_eth1cap $(t-1)$ + 0.51 oc3_eth3cap $(t-1)$ |
| oc3_eth1lab<br>oc3_eth4lab | Oc3_eth1lab = 4729.1 * + 0.35 ** oc3_eth1lab $(t-1)$ + 0.27 oc3_eth4lab $(t-1)$<br>Oc3_eth4lab = 5102.60 * + 0.87 ** oc3_eth1lab $(t-1)$ − 0.19 oc3_eth4lab $(t-1)$ |
| oc3_eth1cap<br>oc3_eth4cap | Oc3_eth1cap = −82.38 + 1.07 * oc3_eth1cap $(t-1)$ + 0.43 * oc3_eth4cap $(t-1)$<br>Oc3_eth4cap = 272.02 − 0.74 oc3_eth1cap $(t-1)$ + 0.56 ** oc3_eth4cap $(t-1)$ |

**Table A2.** VARs of capital and labor earnings by occupation, gender, and citizenship in France (7 Waves 1978–2010). Author's calculation from LIS data (2020).

| Variables Names | Oc1: Managers and Professionals (ISCO 1 & 2)<br>Oc2: Other Skilled Workers (ISCO 3-8, 10)<br>Oc3: Laborers/Elementary (ISCO 9)<br>lab: Average Labor Income<br>lap: Average Capital Income<br>m: Male Population<br>f: Female Population<br>cit1: French Citizenship<br>cit2: French Naturalized Citizens<br>cit3: Non-Citizen Status<br>cit4: African Citizenship Holder<br>cit5: Norther African Citizenship Holder<br>cit6 Europe Citizenship Holder<br>** Significance $\leq$ 0.05<br>* Significance between 0.10 and 0.05 |
|---|---|
| **Variables** | **VARs** |
| oc1_mlab<br>oc1_flab | Oc1_mlab = 10957.93 * − 0.23 oc1_mlab $(t − 1)$ + 1.57 * oc1_flab $(t − 1)$<br>Oc1_flab = 4123.91 * − 0.08 oc1_mlab $(t − 1)$ + 1.12 * oc1_flab $(t − 1)$ |
| oc1_mcap<br>oc1_fcap | Oc1_mcap = 1058.91 * − 1.15 oc1_mcap $(t − 1)$ + 1.92 * oc1_fcap $(t − 1)$<br>Oc1_fcap = 949.85 * − 1.00 oc1_mcap $(t − 1)$ + 1.68 ** oc1_fcap $(t − 1)$ |
| oc1_cit1lab<br>oc1_cit2lab | Oc1_cit1lab = −8355.29 * + 2.38 * oc1_cit1lab $(t − 1)$ − 1.12 * oc1_cit2lab $(t − 1)$<br>Oc1_cit2lab = −12208.73 * + 2.54 * oc1_cit1lab $(t − 1)$ − 1.30 * oc1_cit2lab $(t − 1)$ |
| oc1_cit1cap<br>oc1_cit2cap | Oc1_cit1cap = 6732.47 * − 1.97 * oc1_cit1cap $(t − 1)$ +0.49 * oc1_cit2cap $(t − 1)$<br>Oc1_cit2cap = 12414.87 * − 2.78 * oc1_cit1cap $(t − 1)$ − 1.85 * oc1_cit2cap $(t − 1)$ |
| oc1_cit1lab<br>oc1_cit3lab | Oc1_cit1lab= 7407.12 * +0.96 * oc1_cit1lab $(t − 1)$ − 0.08 * oc1_cit3lab $(t − 1)$<br>Oc1_cit3lab = 28943.92 * − 0.05 * oc1_cit1lab $(t − 1)$ − 0.43 * oc1_cit3lab $(t − 1)$ |
| oc1_cit1cap<br>oc1_cit3cap | insufficient observations |
| oc1_cit1lab<br>oc1_cit4lab | Oc1_cit1lab = 5027.05 * + 0.91 * oc1_cit1lab $(t − 1)$ + 0.11 * oc1_cit4lab $(t − 1)$<br>Oc1_cit4lab = −6861.91 + 0.95 * oc1_cit1lab $(t − 1)$ − 0.59 * oc1_cit4lab $(t − 1)$ |
| oc1_cit1cap<br>oc1_cit4cap | Oc1_cit1cap = 1084.98 * + 0.42 * oc1_cit1cap $(t − 1)$ + 0.76 * oc1_cit4cap $(t − 1)$<br>Oc1_cit4cap = 1451.97 * − 0.30 * oc1_cit1cap $(t − 1)$ − 0.54 * oc1_cit4cap $(t − 1)$ |
| oc1_cit1lab<br>oc1_cit5lab | Oc1_cit1lab = 6963.37 * + 0.88 * oc1_cit1lab $(t − 1)$ + 0.01 oc1_cit5lab $(t − 1)$<br>Oc1_cit5lab = −825.88 +0.59 * oc1_cit1lab $(t − 1)$ − 0.13 oc1_cit5lab $(t − 1)$ |
| oc1_cit1cap<br>oc1_cit5cap | Oc1_cit1cap = 4648.04 * − 0.48 * oc1_cit1cap $(t − 1)$ − 7.97 * oc1_cit5cap $(t − 1)$<br>Oc1_cit5cap = −2122.89 * + 7.96 * oc1_cit1cap $(t − 1)$ +10.12 * oc1_cit5cap $(t − 1)$ |
| oc1_cit1lab<br>oc1_cit6lab | Oc1_cit1lab = 7543.65 * + 0.82 * oc1_cit1lab $(t − 1)$ + 0.07 oc1_cit6lab $(t − 1)$<br>Oc1_cit6lab = −7046.77 +1.10 * oc1_cit1lab $(t − 1)$ − 0.19 oc1_cit6lab $(t − 1)$ |
| oc1_cit1cap<br>oc1_cit6cap | Oc1_cit1cap =763.78 * +0.92 * oc1_cit1cap $(t − 1)$ − 0.41 * oc1_cit6cap $(t − 1)$<br>Oc1_cit6cap = −284.33 +0.90 * oc1_cit1cap $(t − 1)$ − 0.24 oc1_cit6cap $(t − 1)$ |
| oc2_mlab<br>oc2_flab | Oc2_mlab = 4463.35 * + 0.41 oc2_mlab $(t − 1)$ + 0.58 oc2_flab $(t − 1)$<br>Oc2_flab = 2649.01 * + 0.55 ** oc2_mlab $(t − 1)$ +0.18 oc2_flab $(t − 1)$ |
| oc2_mcap<br>oc2_fcap | Oc2_mcap = 233.10 * − 2.21 ** oc2_mcap $(t − 1)$ +2.60 * oc2_fcap $(t − 1)$<br>Oc2_fcap = 301.11 * − 2.64 oc2_mcap $(t − 1)$ +3.02 ** oc2_fcap $(t − 1)$ |
| oc2_cit1lab<br>oc2_cit2lab | Oc2_cit1lab = −2299.50 * − 4.46 * oc2_cit1lab $(t − 1)$ + 6.31 * oc2_cit2lab $(t − 1)$<br>Oc2_cit2lab = 2662.28 * − 0.23 * oc2_cit1lab $(t − 1)$ + 1.16 * oc2_cit2lab $(t − 1)$ |
| oc2_cit1cap<br>oc2_cit2cap | Oc2_cit1cap= 4507.12 * − 2.89 * oc2_cit1cap $(t − 1)$ − 0.86 * oc2_cit2cap $(t − 1)$<br>Oc2_cit2cap = 223.28 * + 0.22 * oc2_cit1cap $(t − 1)$ + 0.24 * oc2_cit2cap $(t − 1)$ |
| oc2_cit1lab<br>oc2_cit3lab | Oc2_cit1lab = 5314.08 * + 0.65 * oc2_cit1ab $(t − 1)$ + 0.16 * oc2_cit3lab $(t − 1)$<br>Oc2_cit3lab = −1203.08 * + 1.47 * oc2_cit1lab $(t − 1)$ − 0.69 * oc2_cit3lab $(t − 1)$ |
| oc2_cit1cap<br>oc2_cit3cap | note: oc2_cit3cap dropped because of collinearity |
| oc2_cit1lab<br>oc2_cit4lab | Oc2_cit1lab = 2513.95 + 0.96 * oc2_cit1lab $(t − 1)$ − 0.03 oc2_cit4lab $(t − 1)$<br>Oc2_cit4lab = 8809.76 + 0.17 oc2_cit1lab $(t − 1)$ − 0.14 oc2_cit4lab $(t − 1)$ |
| oc2_cit1cap<br>oc2_cit4cap | Oc2_cit1cap = 1509.01 * − 0.63 * oc2_cit1cap $(t − 1)$ + 0.42 * oc2_cit4cap $(t − 1)$<br>Oc2_cit4cap = 2000.28 * − 1.73 * oc2_cit1cap $(t − 1)$ − 0.47 * oc2_cit4cap $(t − 1)$ |
| oc2_cit1lab<br>oc2_cit5lab | Oc2_cit1lab = 4838.81 * + 0.78 * oc2_cit1lab $(t − 1)$ − 0.01 oc2_cit5lab $(t − 1)$<br>Oc2_cit5lab = 5555.39 + 0.50 oc2_cit1lab $(t − 1)$ − 0.20 oc2_cit5lab $(t − 1)$ |
| oc2_cit1cap<br>oc2_cit5cap | Oc2_cit1cap = 285.74 ** + 0.59 * oc2_cit1cap $(t − 1)$ + 0.50 oc2_cit5cap $(t − 1)$<br>Oc2_cit5cap = 118.19 + 0.20 oc2_cit1cap $(t − 1)$ − 0.31 oc2_cit5cap $(t − 1)$ |

**Table A2.** *Cont.*

| Variables Names | Oc1: Managers and Professionals (ISCO 1 & 2)<br>Oc2: Other Skilled Workers (ISCO 3-8, 10)<br>Oc3: Laborers/Elementary (ISCO 9)<br>lab: Average Labor Income<br>lap: Average Capital Income<br>m: Male Population<br>f: Female Population<br>cit1: French Citizenship<br>cit2: French Naturalized Citizens<br>cit3: Non-Citizen Status<br>cit4: African Citizenship Holder<br>cit5: Norther African Citizenship Holder<br>cit6 Europe Citizenship Holder<br>** Significance $\leq 0.05$<br>* Significance between 0.10 and 0.05 |
|---|---|
| **Variables** | **VARs** |
| oc2_cit1lab<br>oc2_cit6lab | Oc2_cit1lab = 4951.24 * + 0.71 * oc2_cit1lab $(t-1)$ + 0.07 oc2_cit6lab $(t-1)$<br>Oc2_cit6lab = 2088.77 + 0.79 oc2_cit1lab $(t-1)$ − 0.04 oc2_cit6lab $(t-1)$ |
| oc2_cit1cap<br>oc2_cit6cap | Oc2_cit1cap = 292.98 ** + 1.08 ** oc2_cit1cap $(t-1)$ − 0.41 oc2_cit6cap $(t-1)$<br>Oc2_cit6cap =255.55 + 1.80 * oc2_cit1cap $(t-1)$ − 1.34 * oc2_cit6cap $(t-1)$ |
| oc3_mlab<br>oc3_flab | Oc3_mlab = 6636.90 * − 0.50 oc3_mlab $(t-1)$ +1.20 oc3_flab $(t-1)$<br>Oc3_flab = 4424.80 * − 0.39 oc3_mlab $(t-1)$ + 1.06 oc3_flab $(t-1)$ |
| oc3_mcap<br>oc3_fcap | Oc3_mcap = 169.64 ** − 2.54 oc3_mcap $(t-1)$ + 3.19 ** oc3_fcap $(t-1)$<br>Oc3_fcap = 172.83 * − 2.00 oc3_mcap $(t-1)$ +2.68 ** oc3_fcap $(t-1)$ |
| oc3_cit1lab<br>oc3_cit2lab | Oc3_cit1lab = −1601.95 * +0.08 * oc3_cit1lab $(t-1)$ + 1.20 * oc3_cit2lab $(t-1)$<br>Oc3_cit2lab = 1014.03 * + 0.09 * oc3_cit1lab $(t-1)$ + 0.93 oc3_cit2lab $(t-1)$ |
| oc3_cit1cap<br>oc3_cit2cap | Oc3_cit1cap = 940.38 * − 2.37 * oc3_cit1cap $(t-1)$ +2.55 * oc3_cit2cap $(t-1)$<br>Oc3_cit2cap = 896.50 * − 2.48 oc3_cit1cap $(t-1)$ + 2.22 * oc3_cit2cap $(t-1)$ |
| oc3_cit1lab<br>oc3_cit3lab | Oc3_cit1lab = 4745.62 * + 0.84 * oc3_cit1lab $(t-1)$ − 0.12 oc3_cit3lab $(t-1)$<br>Oc3_cit3lab = −300.11 * + 1.49 * oc3_cit1lab $(t-1)$ − 0.47 oc3_cit3lab $(t-1)$ |
| oc3_cit1cap<br>oc3_cit3cap | insufficient observations |
| oc3_cit1lab<br>oc3_cit4lab | Oc3_cit1lab = 6848.13 + 0.30 oc3_cit1lab $(t-1)$ + 0.13 oc3_cit4lab $(t-1)$<br>Oc3_cit4lab = 9601.59 − 0.27 oc3_cit1lab $(t-1)$ + 0.11 oc3_cit4lab $(t-1)$ |
| oc3_cit1cap<br>oc3_cit4cap | Oc3_cit1cap= 69.84 * + 0.75 * oc3_cit1cap $(t-1)$ + 0.82 * oc3_cit4cap $(t-1)$<br>Oc3_cit4cap = 870.77 * − 1.13 * oc3_cit1cap $(t-1)$ − 0.96 * oc3_cit4cap $(t-1)$ |
| oc3_cit1lab<br>oc3_cit5lab | Oc3_cit1lab = 6123.54 * + 0.44 ** oc3_cit1lab $(t-1)$ + 0.04 oc3_cit5lab $(t-1)$<br>Oc3_cit5lab= 6025.53 + 0.01 oc3_cit1lab $(t-1)$ + 0.41 oc3_cit5lab $(t-1)$ |
| oc3_cit1cap<br>oc3_cit5cap | Oc3_cit1cap = 18839.33 * − 24.06 * oc3_cit1cap $(t-1)$ − 97.78 * oc3_cit5cap $(t-1)$<br>Oc3_cit5cap = −5869.07 * + 8.20* oc3_cit1cap $(t-1)$ + 29.99 * oc3_cit5cap $(t-1)$ |
| oc3_cit1lab<br>oc3_cit6lab | Oc3_cit1lab = 6256.10 * + 0.07 oc3_cit1lab $(t-1)$ + 0.43 oc3_cit6lab $(t-1)$<br>Oc3_cit6lab = 5796.51 * + 0.58 oc3_cit1lab $(t-1)$ − 0.14 oc3_cit6lab $(t-1)$ |
| oc3_cit1cap<br>oc3_cit6cap | Oc3_cit1cap = 134.18 + 0.81 oc3_cit1cap $(t-1)$ + 0.02 oc3_cit6cap $(t-1)$<br>Oc3_cit6cap = 429.75 * − 0.70 oc3_cit1cap $(t-1)$ + 0.58 oc3_cit6cap $(t-1)$ |

**Table A3.** VARs of capital and labor earnings by occupation, gender, and birth in Italy (13 Waves 1986–2016). Author's calculation from LIS data (2020).

| Variables Names | Oc1: Blue-Collar<br>Oc2: Office Worker and Schoolteacher<br>Oc3: Junior/Middle Manager and Professional Occupations<br>Oc4: Senior Managers and White-Collar Workers<br>m: Male<br>f: Female<br>lab: Labor Income<br>cap: Capital Income<br>in: Born in the Country<br>out: Born out the Country<br>** Significance $\leq 0.05$<br>* Significance between 0.10 and 0.05 |
|---|---|
| **Variables** | **VARs** |
| Oc1_mlab<br>oc1_flab | oc1_mlab =2672.60 * + 0.95 * oc1_mlab $(t-1)$ − 0.87 oc1_flab $(t-1)$<br>oc1_flab = 2440.10 * + 0.40 ** oc1_mlab $(t-1)$ + 0.29 oc1_flab $(t-1)$ |
| Oc1_mcap<br>oc1_fcap | Oc1_mcap = 134.97 + 0.32 *oc1_mcap $(t-1)$ + 0.30 oc1_fcap $(t-1)$<br>Oc1_fcap = 52.60+ 0.32 * oc1_mcap $(t-1)$ + 0.57 * oc1_fcap $(t-1)$ |

**Table A3.** *Cont.*

| Variables Names | |
|---|---|
| | Oc1: Blue-Collar<br>Oc2: Office Worker and Schoolteacher<br>Oc3: Junior/Middle Manager and Professional Occupations<br>Oc4: Senior Managers and White-Collar Workers<br>m: Male<br>f: Female<br>lab: Labor Income<br>cap: Capital Income<br>in: Born in the Country<br>out: Born out the Country<br>** Significance ≤0.05<br>* Significance between 0.10 and 0.05 |

| Variables | VARs |
|---|---|
| oc1_inlab<br>oc1_outlab | oc1_inlab = −225.61 + 0.18 oc1_inlab $(t − 1)$ + 1.01 oc1_outlab $(t − 1)$<br>oc1_outlab = 1928.71 + 0.34 oc1_inlab $(t − 1)$ + 0.55 oc1_outlab $(t − 1)$ |
| oc1_incap<br>oc1_outcap | oc1_incap = 221.67 + 0.09 oc1_incap $(t − 1)$ + 0.71 oc1_outcap $(t − 1)$<br>oc1_outcap = 36.21 + 0.34 oc1_incap $(t − 1)$ + 0.27 oc1_outcap $(t − 1)$ |
| oc2_mlab<br>oc2_flab | oc2_mlab = 4097.44 ** + 0.08 oc1_mlab $(t − 1)$ + 0.80 oc2_flab $(t − 1)$<br>oc2_flab = 2384.25 − 0.21 oc2_mlab $(t − 1)$ + 1.16 **oc2_flab $(t − 1)$ |
| oc2_mcap<br>oc2_fcap | oc2_mcap = 401.63 − 0.54 oc2_mcap $(t − 1)$ + 1.06 ** oc2_fcap $(t − 1)$<br>oc2_fcap = 677.23 * − 1.17 ** oc2_mcap $(t − 1)$ + 1.52 * oc2_fcap $(t − 1)$ |
| oc2_inlab<br>oc2_outlab | oc2_inlab = 824.04 + 0.27 oc2_inlab $(t − 1)$ + 0.78 oc2_outlab $(t − 1)$<br>oc2_outlab = 751.78 + 0.28 oc2_inlab $(t − 1)$ + 0.76 oc2_outlab $(t − 1)$ |
| oc2_incap<br>oc2_outcap | oc2_incap = 471.22 + 0.48 oc2_incap $(t − 1)$ + 0.14 oc2_outcap $(t − 1)$<br>oc2_outcap = 664.14 + 0.53 oc2_incap $(t − 1)$ + 0.22 oc2_outcap $(t − 1)$ |
| oc3_mlab<br>oc3_flab | oc3_mlab = 3146.20 + 0.92 * oc3_mlab $(t − 1)$ + 0.15 oc3_flab $(t − 1)$<br>oc3_flab = 8114.84 * + 0.54 * oc3_mlab $(t − 1)$ + 0.03 oc3_flab $(t − 1)$ |
| oc3_mcap<br>oc3_fcap | oc3_mcap = 969.30 ** + 1.19 * oc3_mcap $(t − 1)$ − 0.62 oc3_fcap $(t − 1)$<br>oc3_fcap = 966.40 + 1.02 oc3_mcap $(t − 1)$ − 0.40 oc3_fcap $(t − 1)$ |
| oc3_inlab<br>oc3_outlab | oc3_inlab = 3201.03 + 0.94 * oc3_inlab $(t − 1)$ + 0.77 oc3_outlab $(t − 1)$<br>oc3_outlab = 20.52 + 1.48 * oc3_inlab $(t − 1)$ − 0.64 oc3_outlab $(t − 1)$ |
| oc3_incap<br>oc3_outcap | oc3_incap = 1208.80 ** + 0.16 oc3_incap $(t − 1)$ + 0.33 oc3_outcap $(t − 1)$<br>oc3_outcap = 1701.65 ** − 0.67 oc3_incap $(t − 1)$ + 0.73 **oc3_outcap $(t − 1)$ |
| oc4_mlab<br>oc4_flab | oc4_mlab = 9787.48 + 0.25 oc4_mlab $(t − 1)$ + 0.91 oc4_flab $(t − 1)$<br>oc4_flab = 2378 + 0.34 oc4_mlab $(t − 1)$ + 0.53 oc4_flab $(t − 1)$ |
| oc4_mcap<br>oc4_fcap | oc4_mcap = 3853.93 * + 0.19 oc4_mcap $(t − 1)$ − 0.28 oc4_fcap $(t − 1)$<br>oc4_fcap = 2873.76 * + 0.39 oc4_mcap $(t − 1)$ − 0.27 oc4_fcap $(t − 1)$ |
| oc4_inlab<br>oc4_outlab | oc4_inlab = − 279.29 + 1.60 * oc4_inlab $(t − 1)$ − 0.43 oc4_outlab $(t − 1)$<br>oc4_outlab = − 22010.67 + 3.04 * oc4_inlab $(t − 1)$ − 1.19 ** oc4_outlab $(t − 1)$ |
| oc4_incap<br>oc4_outcap | oc4_incap = 4887.26 * − 0.13 oc4_incap $(t − 1)$ − 0.89 * oc4_outcap $(t − 1)$<br>oc4_outcap = 8252.62 + 0.34 oc4_incap $(t − 1)$ − 0.29 oc4_outcap $(t − 1)$ |

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
