# Peer review of "Sustainable Earnings: How Can Herd Behavior in Financial Accumulation Feed into a Resilient Economic System?"

_sustainability, doi:10.3390/su13115776_

Round 1

Reviewer 1 Report

The formulated research problem is undoubtedly important form the stand point of both society and economy.

The notion that income inequality resultant from unjustified social perceptions instead of the differences in the individual productivity might lead to larger vulnerability of the financial system and emergence of speculative asset bubbles is indeed an interesting research problem.

The Authors justly observe that individual earnings, especially those of the socially privileged groups are unlikely to be allocated to support the green transition instead of speculative investments.

Although the general concept of the policy revision required to address the above issues is valid, the paper seems to lack a precise indication of specific policy instruments that could readily apply to foster the desired change of the existing paradigm.

Therefore, the paper should be enhanced by presenting more detailed policy recommendations.

Language and style of the paper are fine, however minor spell and grammar check is advisable, see e.g. line 266, where the word ‘that’ seems unnecessary.

Author Response

The formulated research problem is undoubtedly important form the stand point of both society and economy.

The notion that income inequality resultant from unjustified social perceptions instead of the differences in the individual productivity might lead to larger vulnerability of the financial system and emergence of speculative asset bubbles is indeed an interesting research problem.

The Authors justly observe that individual earnings, especially those of the socially privileged groups are unlikely to be allocated to support the green transition instead of speculative investments.

Although the general concept of the policy revision required to address the above issues is valid, the paper seems to lack a precise indication of specific policy instruments that could readily apply to foster the desired change of the existing paradigm.

Therefore, the paper should be enhanced by presenting more detailed policy recommendations.

Language and style of the paper are fine, however minor spell and grammar check is advisable, see e.g. line 266, where the word ‘that’ seems unnecessary.

Authors’ Response:

Thank you for your comments.

We have added a section 4.2 on policy recommendations (lines 472 - 512) where the two main recommendations are first in terms of data collection of earnings by group demographic where the demographic element can be passed down the first generation of migrants, i.e. skin colour, which is a major source of discrimination currently missing in two of the three countries looked at. The other recommendation is in terms of a policy tool able to build an open platform of knowledge to show business-as-usual and sustainable earning trends.

We have checked spelling and grammar with the following changes made: “that” in line 266 is removed, other minor changes have been made in lines 30, 32, 73, 76, 98, 188, 358.

Reviewer 2 Report

Though the logical progression of arguments and the overall thesis of the authors is clear, the mathematical apparatus and the proof of evidence is quite lacking (or at leat, they should be described more clearly for the average reader).

Please find hereafter my major concerns:

1) The function Z, which should represent the sum of earnings from capital and labour (lines 107-108), is represented as a function of the demographic groups ("i" and "j") and the occupational group (k), but the occupational group is common to the two demographic groups, so that it plays no role in discriminating between the earnings of the two groups and could be omitted. However, right a line below (line 109), the very same function z is presented as a function of earnings themselves (like a function be a function of itself...). I guess it is just an unfortunate sloppiness in notation, but, if the authors want to use mathematics, they should set it straight.

2) Just a few lines later (line 113) a linear trend over time is assumed for the earnings, which is quite a bold assumption, ignoring economic cycles or anyway possible downturns.

3) In lines 117-122, a first-order autoregressive model is postulated for the profit rates (for capital and labour as well) for individualism. As easily seen, a a first order autoregressive model leads either to a geometrically growing profit rate or to a geometrically decreasing to zero profit rate over time, both of which are quite unrealistic. I understand that the authors describe the model for the short run, but their analysis is conducted over more than 20 years, so that their presentation of individualism is not fair.

4) the vector autoregressive model put forward in lines 129-135 is a very simple model (maybe too simple) the authors postulate for the interaction between the profit rates of the two groups (again, are the authors sure they wish to model the profit rates, i.e. r and w as defined in lines 110-111? or is that again a sloppiness in notation?). However, though its simplicity may be questioned, the authors should provide results as to its goodness-of-fit. Instead, they provide long tables of parameter values for difference combinations of group and different countries, without ever proving that the model adequately describe the experimental data. As a consequence, the conclusions they draw appear not to be adequately supported by their analysis. Any time-series model (be it regressive or autoregressive) should be accompanied by an indication of its goodness of fit, and this makes no exception. In the absence of such indication, just reporting the significance of the coefficient just proves that there is an interaction, but not that the model describes well the reality.

Reviewer 3 Report

The article is well-written and it is a pleasure to read. I think it makes an important contribution. Particularly relevant is comparison of three large European countries.

I have following suggestions:

1) Please expand your discussion of literature in the beginning of article. It is quite limited at the moment but stronger grounding in the literature would be crucial for making the case for your particular political economy perspective. There is plenty of literature on methodological individualism and collective action dilemmas. For instance, literature on bounded rationality, information asymmetry, typology of goods and financialization might be relevant in this context. Your treatment of neo-classical economics seems somewhat naïve considering advances in the literature. 

2) It helps to be clearer about methodology. What concepts are operationalized? Why these methods? Why these variables? What are limitations?

3) Future trajectories are characterized by uncertainty and complexity. Hence, it is worrying that you present your vision in such deterministic way. What possible and plausible other scenarios might emerge from your analysis? What are limitations of your approach concerning future?

Author Response

The article is well-written and it is a pleasure to read. I think it makes an important contribution. Particularly relevant is comparison of three large European countries.

I have following suggestions:

  • Please expand your discussion of literature in the beginning of article. It is quite limited at the moment but stronger grounding in the literature would be crucial for making the case for your particular political economy perspective. There is plenty of literature on methodological individualism and collective action dilemmas. For instance, literature on bounded rationality, information asymmetry, typology of goods and financialization might be relevant in this context. Your treatment of neo-classical economics seems somewhat naïve considering advances in the literature.

Authors’ Response: Thank you for pointing this out. We have expanded on the literature from line 82 to 92 to point out at the political economy perspective adopted in the paper, whereby the context of analysis shapes the rules of legitimacy to access financial resources between social groups. It also shows better in the discussion of the results (lines 244-254; 293-302; 330-337). Equations (1) to (4) are now rewritten as a system of relationships of capital earnings between groups, and relationships of labour earnings between groups. The presentation of individualism is made clearer by making the proposition of a spectrum from individualism at one end to groupism at the other end, with the role of short-term dynamics of earnings over earning gaps being central to the model (lines 141-143 and 167 to 176).  We have also added an example related to the financialization process in the US from line 94 to 97, and tone down the neo-classical critique in line 69, and in the discussion of the results (344-349.

  • It helps to be clearer about methodology. What concepts are operationalized? Why these methods? Why these variables? What are limitations?

Authors’ Response: The concepts being operationalized are made clearer in the new methodological paragraph from line 113 to 121 showing the VAR method used and its justification. The point of the model developed is further explained from line 136 to line 142 and from line 166 to 169. The variables are better displayed in the model as systems of relationships (equations 1 to 4), and the results are exposed more clearly with the new tables 1 to 3. The limitations of the methodology are also described (205-209), which feed back into the new policy recommendations 1 and 2.

  • Future trajectories are characterized by uncertainty and complexity. Hence, it is worrying that you present your vision in such deterministic way. What possible and plausible other scenarios might emerge from your analysis? What are limitations of your approach concerning future?

Authors’ Response: As added in lines 502 to 512, the use of this platform of knowledge by individuals will shape future entitlement rules to financial flows with all its uncertainty and complexity. From a top-down perspective such platform could inform policymakers on labor policies (e.g. job guarantee by demographic group), investment policies (e.g. targeting occupations and sectors supporting the job guarantee), and in-come and wealth policies (e.g. baby bonds and inheritance tax by demo-graphic group). From a bottom-up perspective such platform could inform workers, managers, pensioners, students, men and women on the likely impact their daily financial choices around jobs, savings, and investment by comparing Business-as-usual earning choices and Sustainable Earning choices. There is here no deterministic nature for future flows, rather the future entitlement rules depend on the outcome of individual decisions, whether one chooses to follow his or her own groups’ behavior or not.

Round 2

Reviewer 2 Report

Though the authors have introduced pieces of explanatory text, thus improving the readability of the paper, the model has remained the same and looks as simplistic and as unsupported as in the previous version of the paper. I cannot do but reiterate my previous opinion.

Author Response

The only comment that I can see on the system is from Reviewer 2 who finds the model too simple, the other two reviewers having no further comments.

A simple model doesn’t mean it is a wrong model and it is backed up with all the relevant literature. We have now provided a proof of evidence with three new tables, we have now explained and justified the model clearly (lines 113-121), totally redrafted the writing of the model according to Reviewer 2’s concerns (lines 130-177), described the overall results in line with the model adopted (344-349 and the limitations of the methodology (205-209). It is not clear that Reviewer 2 has engaged with the major revision as no further guidance is provided.

It is now a much improved paper thanks to all of the reviewers’ comments. As it stands, we cannot change the model for being too simple, and it well articulated throughout the paper (as pointed out by reviewers 1 and 2) why it is so.